# Enhancing Cementitious Composites with Functionalized Graphene Oxide-Based Materials: Surface Chemistry and Mechanisms

**DOI:** 10.3390/ijms241310461

**Published:** 2023-06-21

**Authors:** Chien-Yu Huang, Yu-Chien Lin, Johnson H. Y. Chung, Hsien-Yi Chiu, Nai-Lun Yeh, Shing-Jyh Chang, Chia-Hao Chan, Chuan-Chi Shih, Guan-Yu Chen

**Affiliations:** 1Department of Electrical and Computer Engineering, College of Electrical and Computer Engineering, National Yang Ming Chiao Tung University, Hsinchu 300, Taiwan; apple21038526.ee08@nycu.edu.tw; 2Institute of Biomedical Engineering, College of Electrical and Computer Engineering, National Yang Ming Chiao Tung University, Hsinchu 300, Taiwan; s9901231@gmail.com; 3ARC Centre of Excellence for Electromaterials Science, Intelligent Polymer Research Institute, AIIM, Innovation Campus, University of Wollongong, Wollongong, NSW 2500, Australia; johnsonc@uow.edu.au; 4Department of Dermatology, National Taiwan University Hospital Hsin-Chu Branch, Hsinchu 300, Taiwan; extra.owl0430@yahoo.com.tw; 5Department of Medical Research, National Taiwan University Hospital Hsin-Chu Branch, Hsinchu 300, Taiwan; 6Department of Dermatology, National Taiwan University Hospital, Taipei 100, Taiwan; 7Department of Dermatology, College of Medicine, National Taiwan University, Taipei 100, Taiwan; 8Department of Family Medicine, National Taiwan University Hospital Hsin-Chu Branch, Hsinchu 300, Taiwan; yehnailun@gmail.com; 9Department of Obstetrics and Gynecology, Hsinchu Municipal MacKay Children’s Hospital, Hsinchu 300, Taiwan; justine3@ms8.hinet.net (S.-J.C.); chchan0430@gmail.com (C.-H.C.); shihmd0316@gmail.com (C.-C.S.); 10Department of Nursing, Yuanpei University of Medical Technology, Hsinchu 300, Taiwan; 11Department of Biological Science and Technology, National Yang Ming Chiao Tung University, Hsinchu 300, Taiwan

**Keywords:** graphene oxide-based materials, oxygen functional groups, cementitious composites, surface chemistry, nano-reinforced materials

## Abstract

Graphene oxide-based materials (GOBMs) have been widely explored as nano-reinforcements in cementitious composites due to their unique properties. Oxygen-containing functional groups in GOBMs are crucial for enhancing the microstructure of cementitious composites. A better comprehension of their surface chemistry and mechanisms is required to advance the potential applications in cementitious composites of functionalized GOBMs. However, the mechanism by which the oxygen-containing functional groups enhance the response of cementitious composites is still unclear, and controlling the surface chemistry of GOBMs is currently constrained. This review aims to investigate the reactions and mechanisms for functionalized GOBMs as additives incorporated in cement composites. A variety of GOBMs, including graphene oxide (GO), hydroxylated graphene (HO-G), edge-carboxylated graphene (ECG), edge-oxidized graphene oxide (EOGO), reduced graphene oxide (rGO), and GO/silane composite, are discussed with regard to their oxygen functional groups and interactions with the cement microstructure. This review provides insight into the potential benefits of using GOBMs as nano-reinforcements in cementitious composites. A better understanding of the surface chemistry and mechanisms of GOBMs will enable the development of more effective functionalization strategies and open up new possibilities for the design of high-performance cementitious composites.

## 1. Introduction

Cement is a widely used, cost-effective material in the architectural industry, but its pure form is prone to cracking due to brittleness and low tensile strength. Issues such as uneven mixing, high porosity, and susceptibility to environmental damage necessitate the incorporation of protective properties into building materials [1,2,3]. Thus, it is imperative to incorporate protective properties into building materials [4,5]. Appropriate additives are essential to enhance building performance and minimize maintenance costs [1,3,6]. The cement industry is particularly interested in graphene-based nanomaterials with hexagonal carbon networks and oxygen functional groups, which improve the mechanical properties of cement structures [6,7]. Recent studies aim to optimize parameters such as additive dosage, agglomeration state, surface chemistry, oxygen content, defects, and crystal structure to enhance material properties [8,9].

GO contains various oxygen functional groups, such as hydroxyl groups, carboxyl groups, epoxides, carbonyl groups, and ethers, which can be modified through heat or chemical reactions based on the application [10,11,12]. These oxygen-containing functional groups play a crucial role in GO–cement composites, affecting hydration development and microstructure [8]. Gholampour et al. [9] pointed out that balancing the reduction time of GO and the distribution of oxygen functional groups are key parameters for optimizing the physicochemical and mechanical properties of composites. Moreover, Jing et al. [8] noted that GO has the ability to enhance cement hydration and facilitate crystal formation. However, the quantity of oxygen functional groups significantly impacts cement hydration and mechanical strength, especially compressive strength. Furthermore, GOBMs offer electrical conductivity, thermal conductivity, adsorptive properties, and tribological properties, making them potentially useful in signal transmission, monitoring processes, performance, and structural health detection [8,13,14,15,16]. Simple surface chemical modification of GO improves nanomechanical properties, including tribological properties [17,18,19], stiffness, resilience [20,21], and charge transport [19]. Therefore, enhancing the performance of cement with GOBM can have a significant impact on the properties of cement composites.

Previous studies have primarily addressed the strategies for enhancing the properties of cementitious materials by incorporating GO. This review covers the use of GOBM to enhance cementitious materials. We discuss fabrication processes, applications, microstructure, reactions, and interaction mechanisms. Additionally, we introduce a method for controlling oxygen content on GO’s surface as a potential nano-additive for cement composites. Finally, we explore the characteristics and potential applications of different GOBMs (Figure 1).

## 2. Features of GO

### 2.1. Fabrication Process

GO is synthesized from graphite via oxidation and can be traced back to the Brodie method [22]. This technique was improved by the Staudenmaier method, which used a mixture of H_2_SO_4_/HNO_3_ instead of potassium chlorate, and then further improved by the Hummers method [23], which used concentrated sulfuric acid, sodium nitrate, and potassium permanganate to synthesize GO safely and in large quantities (Figure 2a). The chemical reduction approach can determine the carbon-to-oxygen ratio in GO [11]. GO has the potential as an eco-friendly material for cement composites [24]. However, the commonly used Hummers method has drawbacks such as hazardous gas emissions and environmentally unfriendly wastewater. To address these issues, research teams are exploring alternative, safer, and more eco-friendly methods for GO synthesis [25,26,27]. For example, K_2_FeO_4_ is being considered as a replacement for KMnO_4_ in the synthesis process to reduce costs and accelerate the process [28].

### 2.2. Physical and Chemical Properties

#### 2.2.1. Macrostructure

Graphene is a two-dimensional structure with a hexagonal shape consisting of a single layer of carbon atoms. This structure can be wrapped into different dimensional shapes, such as zero-dimensional fullerenes, one-dimensional carbon nanotubes (CNT), and three-dimensional graphite [31]. GO, which is produced using the modified Hummers method, has a size smaller than 100 nm and an average height of approximately 1.4 nm [32]. The GO structure appears irregularly shaped at high magnification using transmission electron microscopy (TEM) [33]. Furthermore, the non-uniform wavy surface of the GO sheet provides thermodynamic stability to the material [34].

#### 2.2.2. Microstructure

Standard methods for analyzing GO-based materials include X-ray photoelectron spectroscopy (XPS), Fourier-transform infrared spectroscopy (FTIR), and Raman spectroscopy. XPS analysis of the C1s in GO reveals three strong peaks at 284.8, 286.8, and 288.1 eV, corresponding to C-C (hydroxyl), C-O (epoxide), and CO (carboxyl), respectively [11,12]. FTIR spectroscopy can be used to analyze the oxygen functional groups of GO, revealing various absorption bands, including O-H stretching vibration at 3000–3700 cm^−1^, which becomes sharper as GO concentration increases [35]. The C=O stretching vibration occurs at ~1714 cm^−1^ but is absent in GO modified with cement. The bands at 1146 cm^−1^ are also observed, corresponding to the epoxy C-O-C bond and the stretching vibration of the C-O bond [36].

The Raman spectrum of GO includes D, G, and 2D bands, which can provide information on the carbon structure of the material. The intensity ratio of each peak can be analyzed to determine the degree of disorder and the number of layers in the material [37]. The D peak near 1350 cm^−1^ is associated with defects in the material, while the G-peak near 1580 cm^−1^ is related to the vibration of sp^2^-hybridized carbon atoms. The 2D peak near 2700 cm^−1^, also known as the G’ peak [37,38], and the I_D_/I_G_ ratio can be used to determine the degree of disorder in GO and the number of defects present in the material [39], which can serve as nucleation sites to accelerate cement hydration [8,40]. On the other hand, the I_2D_/I_G_ ratio is related to the number of graphene layers in the material, with a ratio of 2–3 indicating single-layer graphene, 2 > I_2D_/I_G_ > 1 indicating bilayer graphene, and I_2D_/I_G_ < 1 indicating multi-layer graphene [41].

#### 2.2.3. Surface Electronegativity and π–π Stacking

GO bears a negative charge due to the presence of oxygen functional groups, such as hydroxyl, carboxyl, carbonyl, and epoxide groups, on its surface [11,12,42]. This unique surface chemistry, in comparison with other nanomaterials, like CNT and graphene, reduces van der Waals forces and enhances electrostatic repulsion among the GO sheets, thereby rendering them more hydrophilic [43,44]. The unoxidized regions of the GO surface possess an sp^2^ structure, which facilitates π–π stacking.

The electrical characteristics of GO are intricately linked to its chemical structure. The presence and quantity of oxygen functional groups on GO significantly influence its electrical behavior, enabling it to transition from an insulator to a graphene-like semimetal. This carbon-based material exhibits an sp^2^ and sp^3^ bonding combination, with the electronic structure heavily influenced by π electrons residing on the sp^2^ carbon atoms. Consequently, by manipulating the size and distribution of the sp^2^ region, it becomes feasible to engineer the band gap of GO. This capability renders GO suitable for a wide array of photonic and electronic devices [13,45]. Lee et al. [13] performed morphology and conductance mapping of single-layer GO sheets utilizing conductive atomic force microscopy. The study revealed a direct relationship between the local current in single-layer GO and the presence of hydroxyl, carboxyl, and epoxy groups. It was observed that regions or domains characterized by a greater abundance of sp^2^ honeycomb structures exhibited enhanced conductivity. This notable conductivity increase can be attributed to forming a continuous structure within these regions, enabling a distinct charge carrier migration rate through the graphene framework.

#### 2.2.4. Adsorption Ability

The oxygenated functional groups present in GO result in a hybrid structure with sp^2^ and sp^3^ carbon domains [46]. Due to its 2D layered structure and sp^2^ and sp^3^ domains, GO can be adsorbed in both molecular and ionic forms through various mechanisms, such as electrostatic interactions, π–π interactions, and hydrophobic interactions [47]. Wang et al. [12] conducted a study on the adsorption behavior of GO onto cement using pseudo-first-order and pseudo-second-order rate models. Their findings revealed that GO was more closely aligned with the pseudo-second-order kinetic model, implying the involvement of a chemical reaction during the adsorption process. Furthermore, the Freundlich isotherm model was found to be better suited than the Langmuir model, indicating that the adsorption of GO onto cement occurs in multiple layers [48]. This investigation sheds light on the complex adsorption mechanism of GO onto cement and provides a foundation for future research on this topic.

The initial physical adsorption and chemical interactions between GO and cement significantly impact cement paste hydration and fluidity, as they can capture the divalent cations released by cement hydrates, leading to the acceleration of cement hydration [49]. The larger surface area of GO provides more potential sites for physical and chemical interactions, which can enhance the bonding between the graphene oxide sheets and the material [2,6]. With an increase in GO content in cement composites, the specific surface area, unique 2D structure, and higher surface energy of GO can promote the formation of stronger bonds between cement hydration products and GO sheets. This reduces the fluidity of the cement paste, which can facilitate the wetting of the nanomaterials by attracting more water [50].

## 3. Application of GO Incorporated in Cement Composite

### 3.1. Strength Enhancement

GO is widely used in cement applications due to its remarkable mechanical properties, with intrinsic strength and Young’s modulus of 100 GPa and 1.0 TPa, respectively [2,11,31,36]. Adding GO to cement materials has effectively enhanced mechanical strength and hydration [6]. Cement composites containing GO exhibit improved durability and increased compressive and flexural strengths of approximately 4–70% and 11–42%, respectively [8].

#### 3.1.1. Water Absorption and Cement Hydration

GO exhibits a high water absorption capacity due to its large surface energy and specific surface area (2630 m^2^g^−1^) [11]. The hydration of cement decreases with an increase in the amount of water, and this, coupled with the factors mentioned above, has led to a growing demand for GO–cement composites [51]. Scanning electron microscopy (SEM) observations of the microstructure revealed that, with an increase in the water–cement ratio (w/c) and sufficient free water, the composite structure is loose, while the crystallinity of the prismatic ettringite is high. Large pores and cracks were also observed between the crystals (Figure 2b). With the addition of GO, a higher w/c provides more water molecules, which improves the phenomenon of GO agglomeration, promotes the hydration of cement and the nucleation of hydration products, and reduces the porosity of the cement paste (Figure 2c) [2,8]. Consequently, the structure of the GO-containing composite is more uniform and denser, and its dynamic mechanical properties are improved by the large internal surface area, moderate porosity, and uneven stress distribution of GO [29].

Furthermore, the functional groups present on the surface of GO can undergo a reaction with C_3_S, C_2_S, and C_3_A, forming nucleation sites for the hydrated product [52]. GO has been shown to serve as a template for the arrangement and assembly of cement hydration products into various shapes, such as flowers, rods, needles, and polyhedrons. At a GO content of 0.01–0.04% by weight of cement (bwoc), flower-like crystals were observed to form as the cement hydration products. The flower-like crystals became almost invisible as the curing time increased to 28 days. When the GO content increased to 0.05–0.06% bwoc, polyhedral crystals were observed, with larger polyhedral crystals appearing after 28 days. Moreover, an increase in the GO content resulted in a significant increase in the loss factor and storage modulus of the cement slurry due to the higher degree of internal friction and porosity of the hardened cement paste, which improved the interfacial force within the cement matrix [51]. Likewise, water molecules on the GO surface established water transport channels. The oxygen functional groups of GO play a catalytic role during cement hydration by serving as adsorption sites for water molecules and cement components. Consequently, this interaction considerably expedited the hydration process of cement [53], and the addition of GO improved the dynamic mechanical properties of cement composites, including strength and toughness [29].

#### 3.1.2. Interfacial Transition Zone and Mechanical Properties Improvement

Nanomaterials have a high surface area and chemical properties that promote and densify cement hydration products. They can also serve as scaffolds to support the structure of the interfacial transition zone (ITZ). The chemical properties of nanomaterials, including a high number of oxygen functional groups and strong electrostatic attraction, allow for the forming of chemical bonds with hydrated calcium silicates and cement hydration products. This ability of nanomaterials to establish chemical bonds creates reactive sites that enable subsequent covalent modifications [2,52,54].

The addition of graphene oxide (GO) in cement-based composites can improve the material’s mechanical properties due to the enhanced interfacial interactions between GO and different substrates [55]. Long et al. [50] found that the use of recycled fine aggregate (RFA) mortar without the addition of GO results in the formation of a small number of hydration products that fill the ITZ during cement hardening (Figure 2d). This is due to the high porosity and weak binding of RFA mortars resulting from water absorption. However, when GO is added, the interfacial bonding with RFA is improved through strong covalent bonding between the calcium silicate hydrate (C-S-H) gel and oxygen functional groups of GO (Figure 2e). Additionally, the large surface area of GO provides nucleation sites for cement hydration, improving the mechanical properties of RFA [2]. Furthermore, functionalized GO nanosheets can enhance the interfacial strength and mechanical properties of cement-based nanocomposites, improving both the toughness and Young’s modulus of cement paste [56]. Therefore, GO nanosheets can facilitate and regulate the formation of cement hydrate crystals, which can improve the strength and toughness of cement mortars.

The homogeneity of the dispersion of GO in cementitious materials is crucial for their mechanical durability. However, the mechanical properties of such materials may not continue to improve with the increasing addition of GO beyond a certain point. Li et al. [57] found that while the flexural strength and compressive strength of cement samples initially increased with an increase in GO content, they eventually decreased, with the GO content corresponding to the peak flexural strength differing from that corresponding to the maximum compressive strength. Furthermore, if GO aggregation occurs, its specific surface area decreases, thereby reducing its ability to act as a nucleation site for cement hydration and form a compact microstructure [58]. As a result, the compressive strength of cement paste decreases gradually.

### 3.2. Self-Sensing Capability

Cement composites with self-sensing capabilities are becoming an increasingly popular research area in smart cement materials. By incorporating conductive carbon-based nanomaterials, defects can be detected through changes in the resistance value and piezoresistive properties. Furthermore, the electronegativity of GO allows for the development of self-sensitive cement composites, which can have more economical and versatile construction functions [59]. Achieving high piezoresistive capability and promoting conductivity in GO–cement composites requires the presence of an ideal water content. Consequently, these composites are utilized for crack monitoring and damage identification [8,16].

Certain graphene-based materials possess good electrical conductivity, enabling the formation of conductive networks and effectively decreasing the electrical resistivity of cement, thus making them promising for creating self-monitoring structures [54]. Li et al. [60] indicated that initially, the resistivity of the cement paste increases with the increase in GO content and reaches a maximum when the GO content is 0.02%. However, when the GO content was raised to 0.04%, the resistivity decreased to an estimated value of about 0.003 Ω m, much lower than cement paste. Therefore, the addition of a well-dispersed amount of GO can significantly decrease the resistivity of cement paste, rendering it self-inductive. In accordance with Lavagna et al. [54], the formation of conductive paths in cement is more efficient, and the resistivity is reduced with an appropriate concentration of GO, resulting in a denser and more regular microstructure due to the reduction of cracks and pores. Conversely, the absence of GO or a higher concentration of GO in cement results in a loose and irregular microstructure, thereby disturbing the conductive paths and increasing the resistivity [61]. Hence, it is apparent that the distance between uniformly dispersed conductive fillers is shorter, which allows for the development of a more efficient tunneling conductive path, leading to a strong tunneling effect [62]. Moreover, Rehman et al. [63] reported that slight variations in resistance, rather than changes in resistivity, are the determining factors for self-induction. They further demonstrated that the slope of the fractional change in resistance (FCR) strain (ε) curve increases initially and then decreases upon the incorporation of 0 to 0.15 wt. % of GO content.

According to Guo et al. [64], the piezoresistive capability of GO-incorporated cement composites is significantly influenced by the water content in the cement matrix. The piezoresistive capability of the composite initially increases and subsequently decreases as the water-loss rate of the cement matrix increases. Excessive water content in the cement matrix decreases the composite’s piezoresistive sensitivity because the matrix’s ion conduction is strong and insensitive to deformation. On the other hand, when the water content in the cement matrix is low, the water layer hinders the movement of electrons between different GO sheets, thereby reducing electron conduction and piezoresistive sensitivity. Thus, an appropriate water-to-cement weight ratio and GO content are crucial for GO to enhance the piezoresistive capacity of cement composites effectively.

In summary, GO-based cement composites have the ability to detect defects through resistance changes and piezoresistive characteristics, making them cost-effective and resource-efficient [16,54,59]. However, the effective dispersion of GO within the composites is crucial [60,61,62]. Moreover, the power factor of cementitious substrates is low, and enhancing the thermoelectric properties of cementitious composites is necessary to enable their widespread and economical industrial applications [65].

### 3.3. Challenges and Solutions in Utilizing GO for Cement Composites

Expanding the applicability of GO and cement composites faces two primary challenges: (1) cost limitations and (2) the dispersion of GO in cement, as reported in previous studies [59,66].

Dispersing GO in cement hydration environments containing Na^+^, K^+^, OH^−^, and Ca^2+^ ions is challenging. GO tends to agglomerate in cement when added to the material [2]. The negatively charged GO attracts and reacts with the positively charged Ca^2+^ ions in the cement, forming chemical compounds. The chemical interactions between Ca^2+^ ions and GO in cement primarily involve the attraction of Ca^2+^ ions to the π electrons on the GO surface, which subsequently combine with the functional groups on the edge of GO, resulting in inadequate dispersion of GO [67].

During cement hydration, alkaline compounds are produced that consume the oxygen functional groups on the GO surface, resulting in weakened chemical interactions between GO and cement mortar. This, in turn, restricts the reinforcement capacity [68]. Furthermore, the interaction reduces the fluidity of the cement, leading to increased shrinkage strain as the total volume of the cement decreases [43]. Despite this, GO agglomerates are more electrically insulating than cement paste [60]. Hence, the direct addition of GO is inappropriate for fabricating self-sensing cement composites.

The prevailing methodology entails utilizing ultrasonic oscillation to uniformly disperse GO in the solvent prior to blending with the cement matrix [69]. Despite the use of gentle ultrasonic waves to avert agglomeration and enhance the dispersion of GO, the π–π stacking and van der Waals forces between the GO sheets culminate in their re-agglomeration. Moreover, ultrasonic oscillation is not an apt method for larger concrete blocks as it leads to elevated costs.

## 4. Application of HO-G Incorporated in Cement Composite

### 4.1. Fabrication Process

In the past, HO-G production involved ball-milling graphite flakes with KOH powder [70,71,72]. Nonetheless, this method presents certain drawbacks. Despite the hydroxyl group being present in the resulting HO-G, other oxygen-containing functional groups are also present, thereby limiting the hydroxyl content [73]. Additionally, the hydroxyl groups are predominantly located at the periphery of the graphitic sheets, rendering it arduous to obtain HO-G solutions of high concentration [70].

Sun et al. [73] reported synthesizing HO-G at a high yield. The process involved mixing graphite with H_2_SO_4_ and KMnO_4_, followed by H_2_O_2_ washing to obtain a precursor. Subsequently, the precursor was sheared and emulsified with NaOH dispersion using a high-pressure disrupter. The exclusive presence of hydroxyl groups as the only oxygen-containing group in HO-G was verified through spectroscopic characterization and quantitative halogenation reactions.

In an effort to curtail the usage of combustible and hazardous chemicals such as H_2_SO_4_ and KMnO_4_ in the production of GO and GO derivatives, Yang et al. [70] proposed a comparably straightforward and environmentally sustainable preparation approach. Specifically, HO-G was obtained through ball milling, and the material of choice was porous thermally reduced graphene oxide. The outcomes revealed that the resulting HO-G possessed a greater density of hydroxyl groups located at both the edges of the flat flakes and within the internal nanopores.

### 4.2. Limited Mechanical Enhancement

Despite the significant strengthening effects of GO in various applications, its compatibility with cement composites could be improved [74]. However, the hydrophilicity of HO-G is heightened by the increase in hydroxyl content, rendering it dispersible in cement composites. Adding HO-G has a negligible negative effect on fresh grout in terms of flowability [70]. This is attributable to the broad distribution of hydroxyl groups in HO-G, which enhances interfacial interaction with the C-S-H hydrate. Furthermore, HO-G incorporation refines the pore structure of the cement mortar, inducing greater compaction, accelerating cement hydration, and increasing the yield of hydration products. Additionally, the addition of HO-G significantly curbs the formation and expansion of microcracks through pore-filling and bridging effects. Moreover, it reinforces the resistance of the cement composite to chloride ion penetration (Figure 3a) [70,74].

HO-G can be stably dispersed in cement slurry as a nanofiller without negatively impacting fluidity. However, regarding improving the flexural strength of cementitious composites, HO-G is inferior to GO [74]. The hydroxide or epoxide group weakens the adjacent C-C bond during the transition from sp^2^ to sp^3^ hybridization leading to a structural change from a honeycomb structure to a diamond-like structure. Furthermore, it elucidates that the sp^2^ to sp^3^ transition tends to compromise GO sheets’ structural stability and mechanical properties [14]. Sun et al. [14] investigated the mechanical properties of GO with different functional group densities by molecular dynamics simulations. The findings indicated a notable trend where the fracture stress and Young’s modulus of GO decreased as the density of functional groups decreased. When the same dose of G-OH is added, it exhibits a similar strengthening effect compared to GO in cement composites. This can be attributed to the efficient homogeneous dispersion of 2D nanosheets in the hydrated cement paste, leading to a reduced cross-linking reaction of HO-G with Ca^2+^ in the aqueous solution. Moreover, the weak ligand ability between HO-G and Ca^2+^ results in weaker interfacial forces between HO-G and C-S-H [70]. Pu et al. [74] conducted a study to investigate the effect of combining GO and HO-G on cement composites, considering their respective strengths and weaknesses. The experimental results showed that GO/HO-G composite additives exhibited excellent synergistic enhancement of material properties compared to the addition of GO or HO-G alone. Specifically, at a ratio of 5:5, the compressive strength and resistance to chloride ion migration improved by 40.2% and 21.9%, respectively. The authors suggested that HO-G could assist in the even dispersion of GO in cement mortar due to the similar carbon benzene ring structures of both HO-G and GO. This would fill interfacial gaps and form flower-like patterns of hydrated crystals (Figure 3b).

## 5. Application of ECG Incorporated in Cement Composite

### 5.1. Fabrication Process and Properties

The hydrophilicity of GO nanosheets is partly attributed to the carboxylic groups located at their edges [74]. The production of ECG, which has the potential for carboxylation, is often achieved using ball milling, which is low-cost and high-yield. To generate ECG efficiently, Jeon et al. [75] developed a straightforward and environmentally friendly ball-milling technique that does not require hazardous chemicals. In this study, dry ice was used as a reagent for carboxylation during ball milling, which enabled the capture and storage of CO_2_ gas. Compared to pristine graphite, the surface area of ECG increased by 139 times. In addition, it is noteworthy that the resulting ECG exhibits high dispersibility in protic solvents, such as alkaline aqueous solutions and methanol, as well as in polar aprotic solvents, such as dimethyl sulfoxide and N-methyl-2-pyrrolidone.

### 5.2. Carboxylic Groups of ECG and Interfacial Bond Enhancement

The formation of strong covalent bonds between the carboxylic groups of GO and the hydrated calcium silicates of cement leads to improved adhesion and interfacial bonding between the matrix and reinforcement. Wang et al. [35] have reported that carboxylic groups on GO nanosheets can enhance the adhesion and interfacial bond between the reinforcement and matrix in cement composites. The improved performance is attributed to the chemical reaction between the carboxyl groups at the edges of the GO nanosheets and Ca^2+^ produced during cement hydration. The Ca^2+^ ions act as a connection site to link GO nanosheets with different orientations, forming a 3D network structure (Figure 4a). Additionally, the 3D network structure formed by connecting the GO nanosheets with different orientations via Ca^2+^ connection sites has been observed to embed the hydration products, as revealed by SEM analysis (Figure 4b).

Incorporating GO in cement composites can improve mechanical properties due to the efficient load transfer from the cement matrix to the GO sheet. However, Alkhateb et al. [76] investigated the use of COOH-functionalized graphene nanoplatelets (GNP-COOH) to further enhance the mechanical performance of cement composites. The authors found that GNP-COOH exhibited a relatively high zeta potential in pure water, to further enhance the mechanical performance of cement composites and their efficient dispersion in the cement matrix. Moreover, adding GNP-COOH significantly improved flexural strength, compressive strength, and toughness, with enhancements of 80%, 30%, and 20%, respectively. These improvements may be attributed to carboxylic acid groups on the flakes and the overall sufficient polar group content.

## 6. Application of EOGO Incorporated in Cement Composite

### 6.1. Fabrication Process and Properties

EOGO is a type of GO produced through the ball-milling technique, where graphene powder is subjected to an oxidizing agent with optimal shear and minimal collision force [59,77]. During the ball-milling process, the basal plane of the graphene powder repetitively contacts the steel ball, leading to the removal of oxygen functional groups. In contrast, the active oxygen functional group remains at the edge of the GO (Figure 5a). As a result, the number of reactive oxygen functional groups in EOGO is lower than that obtained through the Hummers method [7]. The active oxygen functional group at the edge of EOGO can be enhanced by increasing the ball-milling time and the amount of oxidizing agent used. The EOGO produced by the ball-milling technique consists of thin flakes with multiple layers [59]. EOGO is manufactured through mechanical and chemical processes, resulting in lower hazardous waste disposal costs. Therefore, it can serve as a promising additive for cement composites, contributing to positive performance effects and potential applications in infrastructure construction.

### 6.2. Mixing Methods

There exist two main mixing techniques for creating composites of EOGO and cement: (1) dry mixing, where EOGO and cement are mixed as dry powders before cement paste formation, and (2) wet mixing, where EOGO solution is subjected to agitation and ultrasound treatment during cement paste formation [78]. Experiments were conducted to compare the efficacy of these two methods by varying the EOGO content, which ranged from 0.01% to 1.0%. Results demonstrated that the consistency of the EOGO–cement composite was superior to that of the control sample. While the compressive strength of the dry-blended EOGO-cement composite can serve as a structural material, the wet-blended composite displayed marginally higher compressive strength. Furthermore, the EOGO–cement composite produced through wet mixing exhibited lower flowability and higher viscosity [59,78]. These findings suggest that the EOGO–cement composite obtained through dry mixing is more practical and offers higher workability, which makes it a better fit for utilization in the concrete industry.

### 6.3. Hydrophilicity of EOGO and Workability Enhancement

The spatial separation between the EOGO flakes and electrostatic repulsion acted as a barrier, preventing the irreversible agglomeration of EOGO and resulting in a high degree of stability of the dispersion in water. Bai et al. [66] reported that EOGO has better dispersibility in solvents than GO, and the dispersed solution appears black, indicating a higher degree of conjugated structures on the basal plane of EOGO than that of GO. The aqueous EOGO solution exhibited excellent stability, as evidenced by the lack of precipitation after a storage period of six months.

Notwithstanding, EOGO displayed partial hydrophobicity due to the distribution of hydrophilic groups at the edges. The SEM-EDS analysis of the 0.05% EOGO-cement composite revealed the presence of needle-like calcium-alumina crystals and amorphous crystallization of C-S-H (Figure 5b) [7,59]. Oxygen groups at the edges of EOGO have been observed to enhance the adsorption of cementitious composites with EOGO. This enhancement promotes the growth of Jennite crystals, ultimately forming dense composites [59,79].

Recent research has explored the effects of incorporating EOGO as a novel admixture into cementitious composites, focusing on its impact on processability, mechanical properties, and adsorption [7,59,79]. EOGO has been shown to enhance the mechanical properties of cement composites by facilitating bridging and filling effects [7]. Specifically, the addition of EOGO at concentrations ranging from 0.01% to 0.1% has been found to increase the workability of cement composites by 26% to 42% [7]. According to Alharbi et al. [79], optimal EOGO content to enhance the strength of cementitious composites was found to be 0.05% and 0.1%, respectively. The compressive strength of the composites was increased by 14.93% and 13.11% on day 7, and by 19.6% and 17% on day 28, respectively. Furthermore, the slump test results indicated that the composite materials containing 0.05% and 0.1% EOGO increased in workability by 31% and 26%, respectively. The probable reasons for the enhanced strength are multifold. Firstly, EOGO in the cement composite is incorporated into the hydration product, thereby reducing microcracking. Secondly, the nanopores of the cement matrix are filled with EOGO, which renders the cement mixture denser. Thirdly, the oxygen groups located at the edges of the EOGO contribute to crystal growth and facilitate nucleation, thus improving the homogeneity of the cement matrix and increasing the compressive strength (Figure 5c,d) [59,79].

## 7. Application of rGO Incorporated in Cement Composite

### 7.1. Fabrication Process

Despite being highly water-soluble and dispersible, GO has an amorphous structure due to its surface’s abundance of oxygen functional groups, resulting in relatively poor mechanical properties compared to pristine graphene [8,9,80]. rGO was thus developed to combine the beneficial properties of GO and pristine graphene, including mechanical, electrical, and thermal properties [81,82,83]. In addition, Kwon et al. [82] showed that the thickness and chemical reduction of both GO and rGO influence their tribological properties. It was observed that the friction behavior of GO is not dependent on its thickness due to the lower friction coefficient and higher adhesion of rGO in comparison to GO. Conversely, the friction exhibited by rGO displays a significant inverse thickness dependence when compared to pristine graphene.

The reduction of GO involves several design factors that need to be considered, including achieving an optimal C/O ratio in the final product, selectively removing specific types of oxygen groups, selecting a suitable green reducing agent, and ensuring the preservation or enhancement of desired physical and chemical properties of GO. These properties encompass mechanical strength, electrical conductivity [84,85], optical properties, and solubility and dispersibility of the resulting nanosheets [81,86]. Various methods have been proposed to decrease the number of oxygen functional groups on the GO surface, including hydrazine solution, plasma methods, and thermal annealing and hydrothermal reduction [31,86,87]. The choice of reduction process can result in different properties for rGO and provide various advantages, such as the low cost and relative ease of chemical reduction. The high-temperature thermal reduction can lead to the exfoliation of GO and the removal of oxygen functional groups through CO or CO_2_ expansion [88]. In contrast to GO, rGO exhibits a multi-layer structure with the thickness in the nanometer scale and diameter on the micron scale and possesses fewer functional groups [31,87]. This property, along with its high electrical conductivity and cost-effectiveness, makes rGO advantageous over GO [89].

### 7.2. Cement Hydration and Mechanical Properties Enhancement

When the cementitious material was mixed with appropriate amounts of rGO, there was a reduction in the internal microcrack density and an increase in the degree of hydration. Moreover, effective bonding with hydrated calcium silicates occurred within the cement matrix [9]. According to Kudžma et al. [50], the addition of GO with a low degree of oxidation (i.e., a C to O ratio of about 4) gradually slowed down the hydration process of cement paste and mortar, but increased their flowability. Incorporating 0.06% GO in the composite led to the most remarkable enhancement in both compressive and flexural strengths, with gains of 22% and 6%, respectively. In another study by Wu et al. [8], the performance of GO and rGO (with a C to O ratio of about 5) as cementitious additives were compared. The results showed that the hydration rate and crystal formation of both types of cement increased (Figure 6a), but the rate with rGO was faster, with a higher total heat of hydration. However, the sample with added GO exhibited better compressive strength.

### 7.3. Thermoelectric Application

Sustainable development is a crucial societal issue, and research on renewable energy and green electricity has gained widespread recognition. In recent years, the thermoelectric properties of composite materials have emerged as a green technology, and they are being increasingly used in road and building construction to address the urban heat island effect and increase the utilization rate of solar energy [90,91].

The thermoelectric effect arises due to the temperature gradient across a thermoelectric material. Under a specific temperature difference, the internal charge carriers of the material diffuse heat towards the lower temperature end, leading to charge accumulation and induction of a thermoelectric voltage [65,92]. The dimensionless figure of merit (ZT) is commonly used to evaluate the thermoelectric behavior, which is expressed as ZT = S2ρ^−1^κ^−1^T, where S is the Seebeck coefficient, ρ is the electrical resistivity, κ is the thermal conductivity, and T is the absolute temperature. Therefore, it can be inferred from the formula that an ideal thermoelectric material should possess a high Seebeck coefficient, low electrical resistivity, and low thermal conductivity, the former two of which are collectively known as thermoelectric power factors [65,92,93]. Cementitious materials can also be employed as thermoelectric materials, producing numerous ions, such as OH^−^, Ca^2+^, K^+^, and Na^+^, during hydration. The thermally induced diffusion of such ions in response to a temperature gradient can facilitate the aforementioned thermoelectric mechanism [65].

The thermal conductivity of GO is approximately 90% lower than that of pristine graphene, and the degree of oxidation of GO can significantly impact its thermal properties. Therefore, researchers have focused on rGO as a potential thermoelectric material [92]. Cui et al. [65] developed a mixed “ionic-electronic” thermoelectric cement-based composite (M-TECC) containing rGO and investigated its thermoelectric behavior under different drying and leaching conditions. Among the composites loaded with 0.00, 0.05, 0.10, and 0.15 wt.% of rGO, the compound with 0.15% rGO exhibited the best thermoelectric performance. Additionally, the Seebeck coefficient of the leaching method was up to two orders of magnitude higher than that of the drying method. The authors suggested that rGO acts as an electronic carrier in this composite material, where electron holes interact with the ions generated by cement hydration, thus amplifying the thermoelectric effect of the composite material.

The presence of oxygen-containing functional groups in GO leads to a decrease in symmetry within its graphene structure, reducing its phonon transport efficiency and causing an acoustic mismatch. Enhancing the sp^2^-hybridized carbon domains and regulating the degree of oxidation of GO at a low level are crucial for improving its thermal properties. Islam et al. [92] conducted experiments to investigate the thermoelectric behavior of thermo-rGO annealed at different temperatures ranging from 100 to 400 °C. The results indicated that rGO annealed at 100 °C exhibited the maximum thermoelectric voltage and Seebeck coefficient. In comparison, rGO annealed at 400 °C showed a dramatic decrease in the Seebeck coefficient, reducing the thermoelectric power factor. The authors concluded that excessive reduction of rGO led to the formation of a graphene-like structure, which decreased the Seebeck coefficient and increased the thermal resistance, thereby negatively affecting the thermoelectric properties.

### 7.4. Importance of Reserved Oxygen Content of rGO

After undergoing the reduction process, rGO experienced a loss of oxygen groups while simultaneously undergoing rearrangement of its graphite structure and lattice defects. Although highly reduced rGO exhibits relatively greater mechanical strength, incorporating this material into composites may only sometimes lead to the highest tensile and compressive strengths. This observation underscores the importance of the oxygen content in the rGO material [9]. Oxygen functional groups and intrinsic defects in GO are crucial in providing nucleation sites for cement hydration. Although rGO has been shown to activate hydration processes, higher amounts of oxygen-containing functional groups are needed to enhance hydration reactions and crystal yields [8]. As such, achieving a balance between the degree of GO reduction and the defects in the graphene structure is imperative.

Gholampour et al. [9] investigated the impact of varying degrees of rGO reduction on the hydration behavior of cementitious composites. The authors found that moderately reduced rGO exhibited the highest degree of cement hydration, whereas highly reduced rGO demonstrated the lowest degree (Figure 6b). The results may be attributed to the abundance of oxidation areas at the interface of the low-reduced rGO, which provides a frictional path for water molecules to penetrate. Conversely, highly reduced rGO contains minimal oxygen functional groups, leading to high hydrophobicity and reduced water molecule capacity.

In addition, specific cement applications, such as concrete pavements, requires special attention to the minimization of frictional resistance, which is essential to increase slip resistance and prevent premature damage [94]. A comprehensive understanding of cementitious composites’ tribological behavior becomes crucial for optimizing their surface properties and enhancing overall performance. The study by Lee et al. [13] indicated that subdomains of GO exhibit low friction (high conductivity) in the sp^2^-rich phase, whereas high friction (low conductivity) is observed in the sp^3^-rich phase. This behavior arises from chemical modifications, such as fluorination, hydrogenation, or oxidation of the single carbon layer, which increase friction by altering the total lateral stiffness and rippling (corrugation) of the potential energy surface. Moreover, Chen et al. [95] showed that due to the non-uniform surface structure of GO, the (~2) GO ultra-thin films with low C/O ratios were susceptible to wear by friction [96]. Zeng et al. [17] compared plasma treatment and thermal reduction methods for obtaining graphene nanosheets with controlled surface wettability and structural defects. Plasma treatment enhanced surface hydrophilicity, but the extended treatment caused structural defects and increased friction. Similarly, thermal reduction led to structural defects and increased friction, with friction decreasing gradually with a longer treatment time.

### 7.5. Fabrication and Mechanism of rGO with Controllable Oxygen Content

Controlling the distribution of functional groups on the surface of rGO is essential due to its unique structural properties. This review proposes a straightforward thermal reduction process to create controllable oxygen-containing functional groups on rGO and discusses the underlying mechanisms.

Traditional methods for producing rGO, such as chemical and high-temperature thermal reduction, have sacrificed oxygen content to enhance GO material properties. However, due to the sub-stable nature of rGO, complex chemical reactions transform the chemical groups on the rGO sheet, limiting the functional material’s stability and reliability [97]. As GO properties improve, with lower sheet resistance and higher chemical reactivity, it is necessary to explore alternative processing techniques to control the GO structure and chemical properties [98,99].

In order to preserve the oxygen content of the GO surface, Kumar et al. [100] proposed a mild thermal annealing technique (50–80 °C) to improve the properties of GO. According to density functional theory (DFT) and material analysis, the process of low-temperature oxygen diffusion on the surface of GO facilitated a phase transition from sp^2^ to sp^3^ domains, leading to the hybridization of the GO surface into significant oxide and graphite domains (Figure 6c). This phenomenon also led to distinct graphite (1–2 nm) and oxygen clusters [101]. The annealed rGO exhibited improved structural stability as a result. In addition, the short-term mild reduction process increased the sp^2^ domain from approximately 45% to approximately 53%, which led to an increase in electron mobility in the graphene plane and electron conductivity by up to four orders of magnitude. This led to a decrease in the resistance of the thin layer by up to four orders of magnitude [101]. Other advantages of this technique include the controllable chemical properties of the GO surface, the absence of chemicals in the reduction reaction, increased visible light absorption, and the potential for large-scale synthesis [98,99].

**Figure 6 ijms-24-10461-f006:**
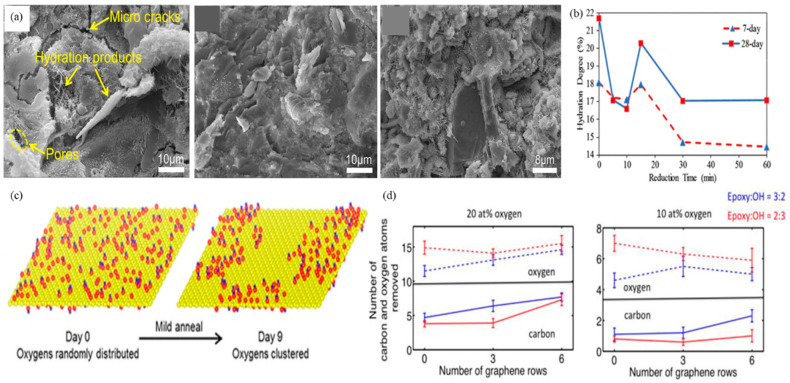
(**a**) (left) After 28 days of curing, the microstructure of cement paste was analyzed; (middle) The control sample showed many pores and microcracks, while the addition of GO resulted in few isolated crystals due to more hydration products; (right) Addition of rGO led to a denser microstructure with fewer pores [8]; Adapted from Ref. [8]. Copyright 2020 Elsevier. (**b**) Moderately reduced rGO was found to have the highest cement hydration, while highly reduced rGO led to a decline in cement hydration, indicating the need to balance the reduction degree of GO with the defects in the graphene structure [9]; Adapted from Ref. [9]. Copyright 2017 American Chemical Society. (**c**) A mild thermal annealing process enhances GO properties by promoting the phase transition of graphitic domains. Oxygen diffusion on the GO surface leads to hybridization of significant oxide and graphite regions, resulting in a mixture of sp^2^ and sp^3^ domains. A diagram featuring carbon (yellow), oxygen (red), and hydrogen (blue) atoms is provided [101]; Adapted from Ref. [101]. Copyright 2017 American Chemical Society. (**d**) GO contains epoxy and hydroxyl groups attached to the edge, which can be removed in high- and low-oxygen environments. Epoxy-rich GO domains lose more carbon and oxygen atoms as the oxygen cluster increases [98]. Adapted from Ref. [98]. Copyright 2016 Elsevier.

Different local distributions of oxygen functional groups are observed on the surface of GO. The GO oxidation region typically contains randomly distributed epoxy and hydroxyl groups attached to the edges. Kumar et al. [98] investigated the impact of different ratios of epoxy to hydroxyl groups (3:2 and 2:3) on the structure and chemical properties of GO during thermal reduction in high- and low-oxygen environments (10 and 20%). The study revealed that the elimination of oxygen and carbon atoms took place in both oxygen environments. However, the reduction process was less significant in the low-oxygen environment. Moreover, an increase in oxygen clustering had a more significant impact on GO domains with higher concentrations of epoxy groups. Consequently, these regions lost more carbon and oxygen atoms (Figure 6d).

## 8. Application of GO/Silane Composite Emulsion Incorporated in Cement Composite

### 8.1. Interfacial Bonding Strength Improvement

Cement-based composites are often integrated with nanomaterials due to their numerous benefits. However, the weak bond strength between the cement hydration products and the aggregate results in high porosity in the ITZ, leading to structural deformation and the failure of concrete reinforcement. This weak component compromises the longevity and durability of concrete, highlighting the importance of enhancing the bond strength between the cement hydration products and nanomaterials at the interface [30,68,102].

Wang et al. [68] prepared functionalized graphene oxide (FGO) using (3-Aminopropyl)triethoxysilane (APTES) and amorphous silica, resulting in the formation of two protective layers. The covalent bond between the amino group of APTES and the oxygen-containing functional group of GO formed the first layer. The second layer effectively isolated GO from an alkaline material using hydrophobic amorphous silica. These layers prevented the carboxyl groups of GO from reacting with alkaline calcium hydroxide formed during cement hydration. The FGO layer improved the bond strength at the ITZ by over 21 times and increased the interfacial adhesion by providing nucleation sites for cement water and product bulk. Furthermore, compared to GO-added samples, FGO-added samples had smaller pore sizes and fewer and smaller calcium alumina crystals at the interface (Figure 7a).

Lee et al. [55] improved the binding strength of carbon fiber/epoxy composites by using silane-functionalized graphene oxide (sGO) fabricated with self-assembled monolayers (SAM). SAM converted the original oxygen-containing GO groups into amines, epoxides, or alkyl groups. This process increased the bond strength of the composite by 29–53% with the addition of sGO. The bonding strength of the sGO-coated samples also increased with the increasing thickness of the sheet, which improved water resistance.

### 8.2. Waterproof Performance and Permeability Resistance

Water easily penetrates concrete through its capillaries due to its hydrophilic nature. This can lead to the transportation of corrosive ions such as chlorides and sulfates, causing damage to the concrete. Moisture within the concrete and its migration are critical factors for the durability of concrete structures in marine engineering. Corrosive ions reacting with cement hydration products can rust the reinforcement, compromising the structural integrity of the concrete [3,102,103]. Two strategies to enhance the longevity and safeguard the structure are incorporating waterproofing materials into the concrete or applying hydrophobic coatings on the surface. Both methods aim to prevent water vapor infiltration and harmful ions, with the former being costlier for larger volumes of concrete [3,4,102].

Nanomaterials and silane-based materials are frequently employed as hydrophobic agents [103]. Graphene-based nanosheets (GNS) possess excellent resistance to permeation, and can be functionalized onto concrete surfaces to form a protective coating, thereby enhancing the durability of the concrete [102]. A composite emulsion of GO and isobutyltriethoxysilane (IBTS) was synthesized to create a waterproof coating for concrete (GS composite emulsion). The application of this emulsion onto concrete resulted in a significant improvement in its waterproofing efficacy.

According to Zhang et al. [104], through the penetration of silane emulsion into the concrete and covalent bonding, a hydrophobic silane layer was formed, converting the hydrophilic concrete surface to hydrophobic. The silane emulsion treatment resulted in a water contact angle of 102.76°, whereas the GS composite emulsion treatment showed a contact angle of 121.53°. The capillary water absorption of the GS composite emulsion-treated concrete was reduced by 93%. Silane coatings exhibit excellent water resistance in foam concrete, a lightweight building component with reduced porosity, and high-water absorption. In a study conducted by Shi et al. [103], the effects of various processing methods, coatings, and soaking on the waterproofing of foam concrete using a GS composite emulsion as the waterproofing material was investigated. The results demonstrated that soaking the foam concrete in the GS composite emulsion resulted in a significantly improved waterproofing effect, with an increase in the thickness of the waterproofing layer by 6.3%.

GO in concrete provides nucleation sites for silane, which enhances the hydrophobic layer’s thickness and reduces micropores [5]. Zhang et al. [104] reported that the inner surface of a sample coated with a silane emulsion displayed a flocculent structure, while one of the samples coated with GS composite emulsion exhibited a flocculent and cluster structure formed by the combination of silane and GO.

The water repellency mechanism of the GS composite emulsion can be elucidated by modeling the water repellency of silane emulsions. When the silane waterproofing material enters the damaged concrete, IBTS undergoes hydrolysis during the synthesis of the GS composite emulsion to form Si-OH bonds. These bonds then condense with different IBTS molecules to form a polymorph structure containing Si-O-Si bonds (Figure 7b). Upon adsorption on foam concrete, the Si-O-Si bond of the polymorph structure and the -OH bond of its hydration product (C-S-H) undergo a dehydration and condensation reaction to form a hydrogen bond, creating a hydrophobic film (Figure 7c). The addition of GO enhances the hydrophobicity further. The Si-OH bond interacts with the -OH bond of GO in a dehydration and condensation reaction. As GO has a high specific surface area and provides numerous active sites, the resulting hydrophobic layer is thicker, more uniform, and denser (Figure 7d) [5,103].

Chen et al. [5] investigated the performance of GS composite emulsion on deteriorated concrete. The results showed that GS composite emulsion-coated concrete had a 75.7% reduction in capillary water absorption coefficient when the concrete was abraded by 5 mm. In contrast, silane emulsion-coated concrete exhibited a 67.4% reduction. When the concrete had a 0.5 mm crack, the capillary water absorption coefficient decreased by 91.4% and 86.1% for GS composite and silane emulsion-coated concrete, respectively. However, the difference in water absorption rates between the two coatings increased with increasing crack width. While both coatings provide water resistance to damaged concrete, GS composite emulsion coating outperforms the silane emulsion coating.

Zhou et al. [4] reported that the GS silane coating is advantageous for safeguarding cement matrices with high porosity. The capillary water absorption coefficient begins to decline when the stress ratio reaches 0.5 *fc* due to the increase in pore size and porosity of the matrix. This facilitates the penetration of the GS composite emulsion into the samples and the formation of waterproof layers. In addition, the authors have proposed two reasons for the microstructural changes in the cement matrix: (1) the reaction of the GS composite emulsion with calcium hydroxide present in the cement to produce hydrated calcium silicates and (2) the hydrolysis of silane materials yielding silanol and hydrated calcium silicate gel, which results in the formation of a hydrophobic layer.

### 8.3. Hydrophobic Coating from Dopamine-Modified GO/Silane Composite Emulsion

According to previous research, incorporating dopamine (DA) into a composite silane emulsion can enhance the protective properties of silane coatings on concrete. Hou et al. [105] found that adding DA to a composite silane emulsion improves the protective properties of silane coatings on concrete. They modified IBTS emulsions using polydopamine-functionalized GO (rGO-PDA) to create a new silane-based nanocomposite coating material. After spraying rGO-PDA-modified silane emulsions on concrete surfaces, the contact angle of the surface reached 140°, and the capillary water absorption was reduced by 90%. Yin et al. [3] noted that DA leads to ion mineralization on concrete surfaces, forming hollow cushion-shaped microstructures that improve interfacial bonding. The authors used DFT to study the adsorption of dopamine molecules with Ca^2+^ and charge density differences and identified three stable structures. The first involved Ca^2+^ adsorption on the nitrogen atom of the amino group (Figure 7e), while the second and third structures had Ca^2+^ adsorbed on oxygen atoms of hydroxyl groups and a carbon ring, respectively (Figure 7f,g). The three structures showed an increase in electron density, suggesting that Ca^2+^ interacts electrostatically with DA at different sites. The second chelate structure represents the most stable structure of the Ca ligand, with the lowest adsorption energy among the three.

## 9. Conclusions and Current Challenges

This review highlights the need for a better understanding of the surface chemistry of GOBMs and their interactions with cement complexes [81,106] (Table 1). Overall, the incorporation of GOBMs into cement composites confers the benefit of improved mechanical strength. This can be attributed to the high surface area and varying proportions of hydrophilic oxygen functional groups that characterize GOBMs, enhancing interfacial bonding with the composites. Moreover, the presence of GOBMs promotes hydration and crystal formation within the composites, thereby reducing pore structure and mitigating the formation of microcracks. However, the surface chemical properties of each type of GOBMs are distinct, necessitating the optimization of a particular surface chemical property for specific materials.

The importance of preserving oxygen content while enhancing other properties for improved performance of cementitious composites is also underscored, along with critical parameters such as size, defects, surface charge, and the number of layers of these graphene-derived materials. Moreover, the widespread use of cement composites with GOBMs also includes:◆Process method: A fundamental issue in the current integration of GOBMs in cement is the tendency towards agglomeration, which results in poor dispersion of the GOBMs within the cement matrix. This can lead to non-uniform mixing of the materials and ultimately, a decrease in the mechanical strength of the resulting blend. Thus, the development of an effective mixing method is imperative for the production of high-performance cement composites.◆Cost: While GOBMs offer numerous benefits not found in traditional building materials, such as nanoscale size and superior mechanical properties, they are comparatively expensive. Consequently, the cost of preparing GOBMs and producing high-quality, stable cement composites incorporating GOBMs and maintaining such building materials must be carefully considered.◆Environment and health: Considering the potential environmental pollution resulting from the production of GOBMs and their composites are crucial in promoting sustainable development. Furthermore, it is crucial to recognize that if GOBMs are released from composites into the environment, they may pose a risk of nano-toxicity through inhalation or ingestion by living organisms.◆Energy: Systematic manufacturing of cement raw materials and subsequent composites requires considerable energy, further increasing cost pressures and hence the need to improve manufacturing efficiency.◆Infrastructure demands: Due to cost considerations, limit the application of cement composites incorporating GOBMs to specific areas or partial segments of a building that require specific properties. For instance, such composites may be employed in bridge piers to confer enhanced curing, improved water resistance, and corrosion resistance, to prolonging their service life. Similarly, in the case of urban skyscrapers, incorporating GOBMs may enable the detection and conversion of heat and electricity, thus optimizing the utilization of waste heat.◆Regulations: As regulations for this innovative building material still need to be completed, manufacturers and end-users face challenges in accurately assessing its quality and safety. Furthermore, the processes of demolishing and recycling abandoned buildings may also require redefinition in light of the unique properties and characteristics of this material.

Despite the need for further research, the rational design of GOBMs holds significant potential as a fabrication strategy for cementitious composites.

## Figures and Tables

**Figure 1 ijms-24-10461-f001:**
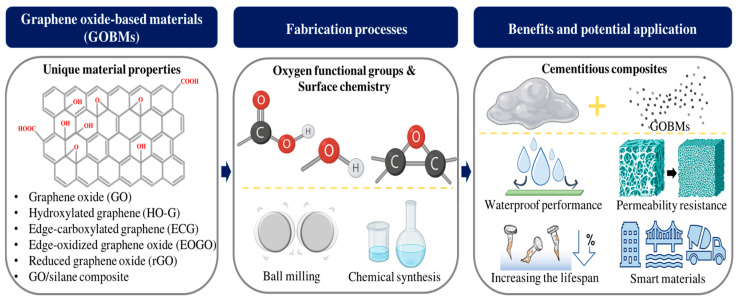
Scope of this review. This review mainly focuses on the various oxygen functional groups present in GOBMs and their integration into cementitious composites for performance improvement. Created with BioRender.com.

**Figure 2 ijms-24-10461-f002:**
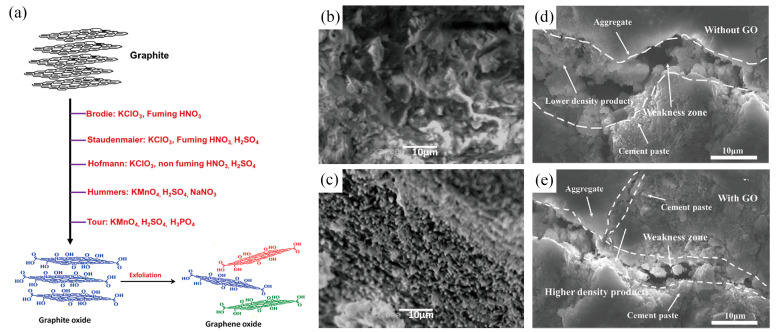
(**a**) Various chemical methods have been employed to synthesize GO from graphite and subsequently exfoliate it, determining the C-to-O ratio in GO [11]. Adapted from Ref. [11]. Copyright 2020, Elsevier. The high surface area, porosity, and water absorption of GO contribute to a more compact composite material structure, thereby improving its mechanical properties. SEM micrographs were taken to observe the microstructure of the cement specimens, where (**b**) represented the specimens without GO and a w/c ratio of 0.35, and (**c**) represented the specimens with GO and a w/c ratio of 0.35 [29]. Adapted from Ref. [29]. Copyright 2019 Elsevier. GO serves as a catalyst for cement hydration products, thereby enhancing interfacial interactions among different substrates. SEM was used to observe the ITZ in (**d**) RFA mortar samples and (**e**) RFA mortar with GO samples at different magnifications after 7 days [30]. Adapted from Ref. [30]. Copyright 2018 Elsevier.

**Figure 3 ijms-24-10461-f003:**
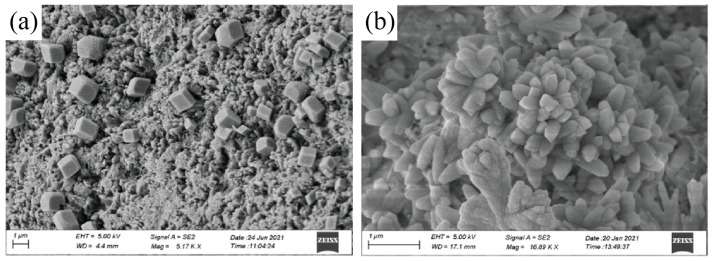
SEM analysis investigated the use of (**a**) HO-G and (**b**) GO/HO-G as nanofillers in cement composites. The images show that the composites containing HO-G and GO/HO-G have a dense texture without porosity or cracks, while GO/HO-G composites exhibit a flower-like pattern of hydrated crystals, indicating a unique interaction with cement hydration products [74]. Adapted from Ref. [74]. Copyright 2022 Royal Society of Chemistry.

**Figure 4 ijms-24-10461-f004:**
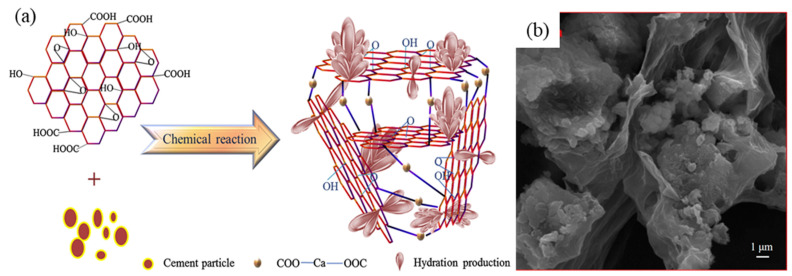
(**a**) The carboxyl group of GO reacts with Ca^2+^ of cement hydration products, forming a strong covalent bond that leads to the formation of a three-dimensional network structure. This structure enhances the interfacial bond between the cement matrix and the reinforcing material [35]; (**b**) SEM images show that the hydrated products are inserted into the 3D network structure of GO nanosheets [35]. Adapted from Ref. [35]. Copyright 2016 Elsevier.

**Figure 5 ijms-24-10461-f005:**
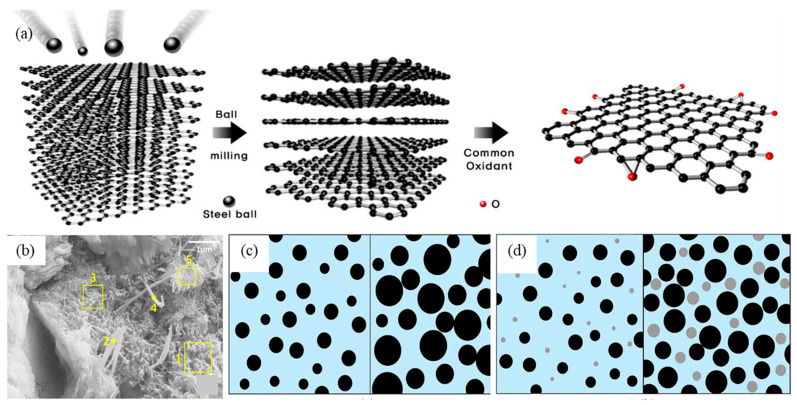
(**a**) Under optimal shear and minimal collision force conditions, graphite powder is oxidized and exfoliated at the edge, resulting in the production of EOGO [77]; Adapted with permission from Ref. [77]. Copyright (2018). MDPI Open access article under the Creative Commons Attribution License. (**b**) The microstructure of cement composites containing EOGO was analyzed by SEM, showing amorphous hydration products in boxes 1, 3, and 5, and needle-like crystals of ettringite at points 2 and 4. A schematic diagram depicted the interaction between EOGO and cement particles, represented by gray and black dots, respectively [59]; (**c**) This figure displays the crystallization process of cement hydration products within the cement pores on day 3 (left) and day 28 (right) in the absence of EOGO [59]; (**d**) When incorporated into cement composites, EOGO can act as an activation point for crystallization, resulting in a more densely packed composite material [59]. Adapted from Ref. [59]. Copyright 2019 Elsevier.

**Figure 7 ijms-24-10461-f007:**
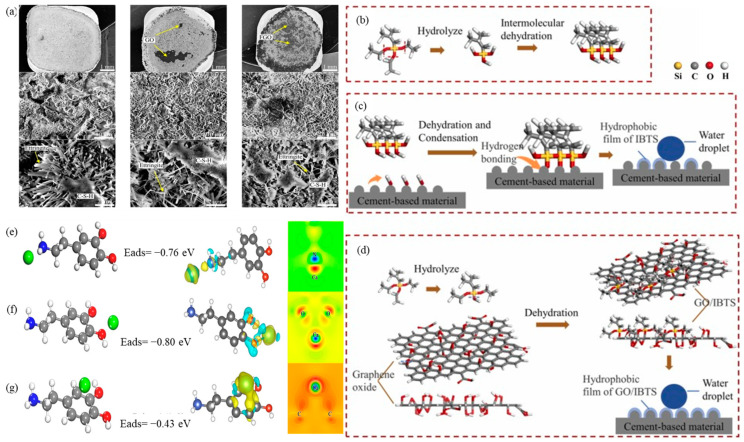
(**a**) SEM images were used to compare the microstructures of (left) the control sample, (middle) GO-containing cement complex, and (right) FGO-containing cement complex. Needle-like crystals (ettringite) were observed in all three samples, but the FGO-containing complex had fewer and smaller crystals. Additionally, the pore size decreased in the order: control sample > GO-containing complex > FGO-containing complex [68]. The scale of the top, middle and bottom columns are 1mm, 10 μm, and 2 μm respectively. Adapted from Ref. [68]. Copyright 2020 John Wiley and Sons. The concrete surface becomes hydrophobic after silane treatment, achieved by the GS composite emulsion mechanism; (**b**) IBTS hydrolysis forms Si-OH and Si-O-Si bonds; (**c**) Si-O-Si bonds undergo dehydration and condensation forming a hydration product with OH bonds, leading to hydrogen bond formation and hydrophobic films, and (**d**) Si-OH bonds interact with -OH groups of GO, resulting in a thicker, denser hydrophobic layer [103]. Adapted from Ref. [103]. Copyright 2022 Elsevier. Dopamine (DA) enhances interfacial bonding on concrete by inducing ion mineralization on its surface. DFT analysis reveals three stable structures of DA molecules on Ca^2+^ with varying adsorption and charge densities. These structures involve Ca^2+^ adsorbed on (**e**) nitrogen, (**f**) oxygen atoms of hydroxyl groups, and (**g**) carbon rings with different bond lengths. The second structure, which closely resembles the Ca-O bond length in CaO crystals, is the most stable due to its low adsorption energy. Electron accumulation is depicted in yellow, while electron depletion is shown in blue [3]. Adapted from Ref. [3]. Copyright 2020 Elsevier.

**Table 1 ijms-24-10461-t001:** The interactions of GOBMs with cement microstructure.

GOBMs	Surface Chemistry	Benefits
GO	hydroxyl groups, carboxyl groups, epoxides, carbonyl groups, and ethers	◆ improve mechanical performance ◆ exhibit high hydrophilicity ◆ enhance hydration ◆ reduce porosity ◆ exhibit limited electrical conductivity ◆ exhibit heat insulation
HO-G	hydroxyl groups	◆ refine pore structure ◆ prevent microcrack formation ◆ increase resistance to chloride ion penetration
ECG	carboxylic groups	◆ enhance mechanical performance ◆ enhance adhesion and interfacial bonding
EOGO	the remaining active oxygen functional group at the edge of GO	◆ improve mechanical performance ◆ increase interfacial bonding ◆ enhance the adsorption of cementitious composites ◆ promote crystals and dense composites growth
rGO	the remaining active oxygen functional group	◆ improve mechanical performance ◆ enhance hydration ◆ exhibit limited hydrophilicity ◆ decrease microcrack density ◆ demonstrate higher hydration heat ◆ a promising thermoelectric material ◆ improve electrical conductivity ◆ exhibit low resistivity
GO/silane composite	silane-functionalized composite	◆ increase interfacial bonding ◆ refine the microstructure ◆ excellent water-repellent property ◆ biofouling-resistant hydrophobic surfaces

## Data Availability

Data are contained within the article. These data can be made available upon request.

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
