# Peer review of "Enhancing Cementitious Composites with Functionalized Graphene Oxide-Based Materials: Surface Chemistry and Mechanisms"

_ijms, 2023, doi:10.3390/ijms241310461_

Round 1

Reviewer 1 Report

The Report of the manuscript  ijms-2432927 is attached.

The Report of the manuscript  ijms-2432927 is attached.

Author Response

Response to Reviewer 1 Comments

The manuscript is well written - clear derivations and explanations.

Before the Editor decides, I suggest that the authors must consider the following corrections:

Point 1: 1. The title should be rewritten to be friendly.

Response 1:

We are very grateful for the Reviewer’s suggestions. We have revised the title from “Impact of oxygen functional groups in graphene oxide-based materials on cementitious composites” to “Enhancing Cementitious Composites with Functionalized Graphene Oxide-Based Materials: Surface Chemistry and Mechanisms” to make it to read and understand. The revisions made in the manuscript have also been highlighted in red and marked up using the “Track Changes” function for easy evaluation.

Point 2: The Introduction section must be more concise.

Response 2:

We are very grateful for the Reviewer’s suggestions. We have revised the Introduction to make it more concise. The revisions made in the manuscript have also been highlighted in red and marked up using the “Track Changes” function for easy evaluation.

Point 3: Add the DOI at References where it is possible.

Response 3:

Thanks to the Reviewer's suggestion, we have carefully checked and re-added all the citations in the manuscript's reference.

Point 4: What are the advantages of the proposed new model?

Response 4:

The benefits of the new model include the combination of GO and rGO material properties, and it is made by gentle thermal annealing, which is a relatively energy-efficient and environmentally friendly method. In addition, the oxygen functional group distribution at the interface is controllable and can be adjusted for experimental purposes.

Point 5: Improve your English and check the spelling of certain words in the body text of the paper.

Response 5:

Thanks to the Reviewer's suggestion, we carefully checked and revised all the contents in the manuscript. The revisions made in the manuscript have also been highlighted in red and marked up using the “Track Changes” function for easy evaluation.

Point 6: The literature survey must be improved by adding some relevant references using the same procedures, for example: - Mechanics of Elastic Composites, Chapman & Hall/ CRC Press, U.S.A, 708 pp., (2004); Electrochemical Behavior of Reduced Graphene Oxide Supported Germanium Oxide, Germanium Nitride, and Germanium Phosphide as Lithium-Ion Battery Anodes Obtained from Highly Soluble Germanium Oxide, Int. J. Mol. Sci. 2023, 24(7), 6860; https://doi.org/10.3390/ijms24076860.

Response 6:

Thanks to the Reviewer's suggestion, the citation was added on page 12-line 520. The revisions made in the manuscript have also been highlighted in red and marked up using the “Track Changes” function for easy evaluation.

Reviewer 2 Report

​I really cannot understand what is the purpose of this review if you only write a few sentences? Please modify this manuscript in the next version, it is not suitable to publish with such forms, it is very difficult to understand.

1) Why in Figure 2-b agglomerated, discuses are obligated?
2) Change 3.3. current challenges titer when it is not useful.
3) The quality of SEM images are low, they could be modified.

The quality of English grammar must be modified.

Author Response

Response to Reviewer 2 Comments

​I really cannot understand what is the purpose of this review if you only write a few sentences? Please modify this manuscript in the next version, it is not suitable to publish with such forms, it is very difficult to understand.

1) Why in Figure 2-b agglomerated, discuses are obligated?
2) Change 3.3. current challenges titer when it is not useful.
3) The quality of SEM images are low, they could be modified.

Response :

We sincerely appreciate your time and effort in reviewing our manuscript, and your constructive comments that will undeniably enhance its quality. We have carefully considered each of your points and addressed them as detailed below:

Your comment about the brevity of the review section caught our attention, and we do apologize if it seemed incomplete. Despite the total manuscript length exceeding 24 pages, we understand that some sections may have seemed underdeveloped. Therefore, we have expanded the discussions and further clarified the purpose of this review to present a more comprehensive understanding.

Concerning Figure 2-b, we have added detailed explanations about the reason and implications of agglomeration in the revised manuscript. We have reconsidered the title of section 3.3 and adjusted it to more accurately reflect the content following your suggestion. We believe this will aid the reader's comprehension of our challenges and findings. Regarding the quality of the SEM images. We have replaced them with higher resolution images if needed. If there are still any issues, we would appreciate further guidance on which specific aspects need to be improved.

We have revised the manuscript extensively to address your concerns and have marked these changes in red using the "Track Changes" function for your convenience.

Once again, we sincerely thank you for your insightful comments. Your feedback is instrumental in enhancing the quality and comprehensibility of our work.

Reviewer 3 Report

In the manuscript, the reactions and mechanisms for functionalized graphene oxide for the application of cement composites are discussed and reviewed. The paper has the focus of the role of the oxygen-containing functional groups that enhance the response of cementitious

composites, and controlling the surface chemistry of GOBMs are intensively discussed. A variety of GOBMs, including graphene oxide (GO), hy- droxylated graphene (HO-G), edge-carboxylated graphene (ECG), edge-oxidized graphene oxide (EOGO), reduced graphene oxide (rGO), and GO/silane composite, are discussed with regard to their oxygen functional groups and interactions with the cement microstructure. I find that the paper deals with the important and timely subject, and in general, it is written clearly and also organized well. Therefore, I suggest the paper be accepted after the proper revision. Below are my comments and suggestions that need to be addressed.

1.     The various treatment on the graphene changes the surface chemistry as well as the nanomechanical and tribological properties of GOBMs. Due to the relevance of the subject, I suggest the following papers about nanomechanical and tribological properties of chemically modified graphene can be mentioned and cited. [For example, J. T. Robinson et al. Nano Letters, 8, 3441 (2008); J. H. Ko et al. Tribology Letters, 50, 137-144 (2013); B. Sharma et al. Polymer Testing, 70, 458-466 (2018); J. Y. Park et al. Advanced Materials Interfaces 1, 1300089 (2014)]

2.     The ratio between sp2 and sp3 of these GOBMs is the important factor to determine the nanomechanical properties as reported earlier [H. Lee et al. Nanoscale 8, 4063 (2016); Y. Sun et al. RSC Adv., 10, 29610-29617 (2020)]. This point can be included in the paper.

3.     At page 12, it was mentioned that despite being highly water-soluble and dispersible, GO has an amorphous structure due to the abundance of oxygen functional groups at its surface, resulting in relatively poor mechanical properties compared to pristine graphene. rGO was thus developed to combine the beneficial properties of both GO and pristine graphene. Indeed, there are many reports showing the mechanical and tribological properties of rGO, in comparison with those of GO [ For example, A. Smith et al. Nano Materials Science, 1, 31-47 (2019): Kwon et al. J. Phys. Chem. B 122, 543−547 (2018) ] In this regard, the more detailed comparison between GO and rGO can be made in the table 1.

Moderate editing of English language required

Author Response

Response to Reviewer 3 Comments

In the manuscript, the reactions and mechanisms for functionalized graphene oxide for the application of cement composites are discussed and reviewed. The paper has the focus of the role of the oxygen-containing functional groups that enhance the response of cementitious composites, and controlling the surface chemistry of GOBMs are intensively discussed. A variety of GOBMs, including graphene oxide (GO), hy- droxylated graphene (HO-G), edge-carboxylated graphene (ECG), edge-oxidized graphene oxide (EOGO), reduced graphene oxide (rGO), and GO/silane composite, are discussed with regard to their oxygen functional groups and interactions with the cement microstructure. I find that the paper deals with an important and timely subject, and in general, it is written clearly and also organized well. Therefore, I suggest the paper be accepted after the proper revision. Below are my comments and suggestions that need to be addressed.

Point 1: The various treatment on the graphene changes the surface chemistry as well as the nanomechanical and tribological properties of GOBMs. Due to the relevance of the subject, I suggest the following papers about nanomechanical and tribological properties of chemically modified graphene can be mentioned and cited. [For example, J. T. Robinson et al. Nano Letters, 8, 3441 (2008); J. H. Ko et al. Tribology Letters, 50, 137-144 (2013); B. Sharma et al. Polymer Testing, 70, 458-466 (2018); J. Y. Park et al. Advanced Materials Interfaces 1, 1300089 (2014)]

Response 1:

We appreciate the Reviewer’s suggestions. The citation was added on page 2- line 61~66 and line 69~71, page 5-line 209~213, page 8-line 368~371. The revisions made in the manuscript have also been highlighted in red and marked up using the “Track Changes” function for easy evaluation.

Point 2: The ratio between sp2 and sp3 of these GOBMs is the important factor to determine the nanomechanical properties as reported earlier [H. Lee et al. Nanoscale 8, 4063 (2016); Y. Sun et al. RSC Adv., 10, 29610-29617 (2020)]. This point can be included in the paper.

Response 2:

We appreciate the Reviewer’s suggestions. The citation was added on page 2- line 68~69, page 4-line 130~152, page 8- line 364~368 and page 14-line 609~625. The revisions made in the manuscript have also been highlighted in red and marked up using the “Track Changes” function for easy evaluation.

Point 3: At page 12, it was mentioned that despite being highly water-soluble and dispersible, GO has an amorphous structure due to the abundance of oxygen functional groups at its surface, resulting in relatively poor mechanical properties compared to pristine graphene. rGO was thus developed to combine the beneficial properties of both GO and pristine graphene. Indeed, there are many reports showing the mechanical and tribological properties of rGO, in comparison with those of GO [ For example, A. Smith et al. Nano Materials Science, 1, 31-47 (2019): Kwon et al. J. Phys. Chem. B 122, 543−547 (2018) ] In this regard, the more detailed comparison between GO and rGO can be made in the table 1.

Response 3:

We really appreciate the Reviewer’s suggestions. We have made a more detailed comparison between GO and rGO in Table 1, and the reference was added on page 18-line 803. The revisions made in the manuscript have also been highlighted in red and marked up using the “Track Changes” function for easy evaluation.

Round 2

Reviewer 2 Report

This version is better for publishing, 

This version is better for publishing, 

Reviewer 3 Report

After revision, the quality and the presentation of the paper have been improved significantly. Therefore, I suggest the paper be accepted as it is.